# Biosynthesis of Sesquiterpenes in Basidiomycetes: A Review

**DOI:** 10.3390/jof8090913

**Published:** 2022-08-28

**Authors:** Jiajun Wu, Xiaoran Yang, Yingce Duan, Pengchao Wang, Jianzhao Qi, Jin-Ming Gao, Chengwei Liu

**Affiliations:** 1Key Laboratory for Enzyme and Enzyme-like Material Engineering of Heilongjiang, College of Life Science, Northeast Forestry University, Harbin 150040, China; 2Shaanxi Key Laboratory of Natural Products & Chemical Biology, College of Chemistry & Pharmacy, Northwest A&F University, Yangling 712100, China

**Keywords:** basidiomycetes, sesquiterpene, biosynthesis, sesquiterpene synthase

## Abstract

Sesquiterpenes are common small-molecule natural products with a wide range of promising applications and are biosynthesized by sesquiterpene synthase (STS). Basidiomycetes are valuable and important biological resources. To date, hundreds of related sesquiterpenoids have been discovered in basidiomycetes, and the biosynthetic pathways of some of these compounds have been elucidated. This review summarizes 122 STSs and 2 fusion enzymes STSs identified from 26 species of basidiomycetes over the past 20 years. The biological functions of enzymes and compound structures are described, and related research is discussed.

## 1. Introduction

Fungi are widely distributed in various ecosystems of the Earth. Based on high-throughput sequencing methods, approximately 5.1 million species of fungi exist in nature, but only approximately 100,000 species have been discovered [1]. Basidiomycota (commonly known as basidiomycetes) is one of the major phyla of the fungal kingdom, with more than 31,000 species identified [2]. Basidiomycetes are divided into three subphyla: rusts (Puccinomycotina), smuts (Ustilagomycotina), and mushrooms (Agaricomycotina), with several taxonomic ranks below them. Sesquiterpenes are among the most structurally diverse natural products and have many applications in various industries. They contain C15 polymers composed of three isoprene units and derivatives with diverse chemical skeletons. Fungi are rich in sesquiterpenoid natural products, many of which have good biological activities, including antibacterial, antifungal, anti-inflammatory, antitumor, vascular-relaxing, immunosuppressant, and cytotoxic activities. They can be used as lead compounds for new drugs [3,4,5,6,7,8,9,10]; especially in basidiomycetes, sesquiterpenes have various pharmacological activities [11].

Basidiomycetes often produce large fruiting bodies to disperse spores; however, these fruiting bodies are constantly threatened by other organisms that feed on them [12]. As a result, basidiomycetes have evolved a number of protective strategies against threats from other organisms, one of which is the production of toxins. Basidiomycetes produce toxic sesquiterpenes, mainly as protoilludane skeleton, to protect against predators [11]. In addition, basidiomycetes often form symbiotic relationships with roots and their hosts, providing plant hormones [13,14]. For example, basidiomycetes in the genus *Lactarius* produce modified lactarane and protoilludane-derived sesquiterpenes that promote plant growth [15,16,17]. Sesquiterpenes isolated from basidiomycetes also exhibit pharmacological activity. For instance, hydroxymethylacylfulvene (HMAF) is a semisynthetic antitumor agent based on the naturally occurring illudin S from the mushroom *Omphalotus olearius* [18]. It is currently in human clinical trials because of its anti-cancer properties [19,20]. Phellinignin A and 11,12-epoxy-12β-hydroxy-1-tremulen-5-one isolated from the genus *Phellinus igniarius* showed high cytotoxicity to HL-60, SMMC-7721, and SW480 cancer cells [21]. 10β,12-Dihydroxy-tremulene isolated from *Phellinus igniarius* showed good vasodilatory activity in the experiment [9].

So far, approximately one thousand sesquiterpenoids have been reportedly obtained from basidiomycetes (Appendix A) [22,23]. Drimanes, protoilludanes, illudanes, hirstutanes, cadinanes, and tremulanes sesquiterpene skeletons are the main skeleton types of sesquiterpenoids in basidiomycetes, comprising approximately 60% of the total population (Figure 1, Appendix A. There are 79 genera of basidiomycetes that produce sesquiterpenes, and *Lactarius*, *Xylaria*, *Armillaria*, *Phellinus*, *Granulobasidium*, and *Conocybe* are the main sources (Appendix A). According to the reported genomic data, the average number of sesquiterpene genes per strain in basidiomycetes is 12, which is much higher than the average 3.5 sesquiterpene genes in ascomycetes [24]. These results suggest that basidiomycetes produce more sesquiterpenes. However, most of the compounds produced by these potentially functional genes are unknown and require further clarification.

## 2. Cyclization Mode of STSs

The sesquiterpene biosynthetic pathway is divided into two steps [25]. The first step is a coupling reaction that connects isoprene precursors, dimethylallyl dipyrophosphate (DMAPP) and isoprenyl dipyrophosphate (IPP), from geranyl pyrophosphate (GPP) in a head-to-tail manner, and then condenses with another molecule of IPP to generate farnesyl pyrophosphate (FPP), which is a sesquiterpenoid biosynthetic precursor and substrate for STS [26]. As the second step, FPP generates different sesquiterpene carbon skeletons through irregular coupling reactions. Typical STS contains conserved D(D/E)XXD and NSE/DTE motifs, and these amino acid residues play important roles in coordinating the stabilization of divalent metal ions at the active site for defocusing the catalytic reaction of phosphoric acid. Cyclization is initiated by the metal-ion-induced departure of inorganic pyrophosphate (PPi) to form allyl cations, facilitating the structural shift and catalyzing cyclization closure [27,28]. For cyclic sesquiterpenes, this step can be further divided into two. FPP undergoes one or more cyclizations to form intermediates, which are then converted to sesquiterpene skeletal end-products under the action of STS. The reaction mechanism is divided into four categories [29,30,31] (Figure 2 and Figure 3)—Clade I: After (2*E*,6*E*)-FPP is deionized by pyrophosphate, it electrophilically attacks the double bond at the other end and forms a 10-membered ring carbon-positive intermediate E, E-germacradienyl cation through a 1,10 cyclization reaction; Clade II: FPP is first ionized and isomerized to form (3*R*)-nerolidyl diphosphate ((3*R*)-NPP), deionized by pyrophosphate, and electrophilically attacks the double bond to form a 10-membered ring of the carbon-positive intermediate *Z*, *E*-germacradienyl cation through 1,10 cyclization; Clade III: (2*E*,6*E*)-FPP removes pyrophosphate ionization, electrophilically attacks the double bond at the other end, and undergoes 1,11 cyclization reaction 11-membered ring carbocation intermediate trans-humulyl cation; Clade IV: After (3*R*)-NPP is deionized by pyrophosphate, it electrophilically attacks the double bond and forms a 6-membered ring carbocation intermediate (6*R*)-β-bisabolol cation through a 1,6 cyclization reaction.

Over the past 20 years, 122 STSs and 2 fusion enzymes STSs have been discovered and identified from 26 species of basidiomycetes (Appendix A), which are responsible for the biosynthesis of hundreds of sesquiterpenes in four ways. The various STSs and their catalytic production of sesquiterpenes are summarized and discussed in this review.

## 3. STSs in Basidiomycota

### 3.1. Agaricales

Agaricales is the largest mushroom-forming flora, comprising more than 400 genera and 13,000 species [32]. To date, 10 species have been experimentally identified with 58 different STSs.

#### 3.1.1. *Macrolepiota albuminosa*

*Macrolepiota albuminosa* (*Termitomyces albuminosus*) is a special mushroom in China that belongs to the Agaricaceae family. Bioinformatic analysis of the genome revealed the presence of 22 terpene synthases [33], 3 of which (*STC4*, *STC9*, and *STC15*) were identified as STSs and heterologously expressed by *Escherichia coli* [34]. Using FPP as a precursor, *STC4* synthesized intermedeol (**1**) via C1,10 cyclization, which in turn enabled germacrene D-4-ol (**2**) synthesization by *STC15*. Through C1,6 cyclization, γ-cadinene (**3**) was synthesized by *STC9* with NPP as a substrate.

#### 3.1.2. *Coprinopsis cinerea*

*Coprinopsis cinerea* belongs to the Psathyrellaceae family. Six STSs (*Cop1–6*) have been identified in this fungus [29] and heterologously expressed in *Saccharomyces cerevisiae* and *E. coli*. With FPP as the precursor, *Cop1* and *Cop2* synthesized germacrene A (**4**) by C1,10 cyclization, and *Cop3* synthesized α-muurolene (**5**). Using NPP as the precursor, *Cop4* synthesized δ-cadinene (**6**) by C1,10 cyclization, and *Cop6* synthesized α-cuprenene (**7**) by C1,6 cyclization [35]. *Cop5* cannot be functionally expressed in either system.

#### 3.1.3. *Taiwanofungus camphoratus*

*Taiwanofungus camphoratus* (*Antrodia cinnamomea*) belongs to the mushroom family Fomitopsidaceae and is a rare medicinal fungus found in Taiwan, China. A total of 10 terpene synthases (*AcTPS1–7*, *9–11*) have been identified in it and they are heterologously expressed in *E.coli*, among which three were identified as STSs (*AcTPS4*, *AcTPS**5*, and *AcTPS**9*) [31]. T-cadinol (**8**) was synthesized by C1,10 cyclization of *AcTPS5* with FPP as a substrate, and cubebol (**9**) and zonarene (**10**) were synthesized from *AcTPS9* and *AcTPS4*, respectively, via C1,10 cyclization with NPP as a substrate.

#### 3.1.4. *Clitopilus pseudo-pinsitus*

*Clitopilus pseudo-pinsitus*, belonging to the family Entolomataceae, currently has 18 related STSs (*CpSTS1–18*) recorded [36]. Apart from the lack of conserved *CpSTS10* sequences, the remaining 17 were heterologously expressed by *Aspergillus oryzae*, and *CpSTS15* was found to be inactive. The biosynthesis of the remaining STSs can be summarized as follows: Δ^6^-protoilludene (**11**) was synthesized by *CpSTS4* using FPP as a precursor and C1,11 cyclization. Sterpurene (**12**), pentalenene (**13**), and α-farnesene (**14**) were synthesized from *CpSTS1*, *CpSTS6*, and *CpSTS7*, respectively. After C1,10 cyclization, δ-cadinene (**6**) was synthesized from *CpSTS2*, aristolene (**15**) was synthesized from *CpSTS16*, and alloaromadendrene (**16**) and 9-alloaromadendrene (**17**) were synthesized from *CpSTS8* and *CpSTS11*, respectively. *CpSTS9* and *CpSTS12* synthesized virifloridol (**18**), and *CpSTS13* synthesized ledene (**19**) with NPP as the substrate. *CpSTS3* synthesized α-muurolene (**5**) and δ-cadinol (**20**). *CpSTS5* synthesized α-muurolene (**5**) with FPP as the substrate. Through C1,6 cyclization, using NPP as a precursor, *CpSTS14* synthesized β-elemene (**21**) and β-farnesene (**22**), *CpSTS17* synthesized β-caryophyllene (**23**), and *CpSTS18* synthesized γ-cadinene (**3**).

#### 3.1.5. *Hypholoma fasciculare*

*Hypholoma fasciculare* is a clustered fungus belonging to the family Strophariaceae. A total of 17 STSs have been previously identified in their genome using bioinformatic methods [37], of which 4 (*Hfas94a*, *Hfas94b*, *Hfas255*, and *Hfas344*) were heterologously expressed in *A. oryzae*. Using FPP as a precursor, *Hfas94a* and *Hfas94b* mainly synthesized α-humulene (**24**) through C1,11 cyclization. *Hfas255* did not produce any products, and *Hfas344* synthesized an oxidized sesquiterpene with spectral data similar to that of β-caryophyllene.

#### 3.1.6. *Hypholoma lateritium*

*Hypholoma lateritium* (*Hypholoma sublateritium*) belongs to the same genus as *H. fasciculare* and is widely distributed in China. Only one STS (*Hypsu1_138665*) has been identified in this mushroom [38]. Using *E. coli* heterologous expression with FPP as the precursor, *Hypsu1_138665* was cyclized by C1,11 cyclization to synthesize Δ^6^-protoilludene (**11**).

#### 3.1.7. *Cyclocybe aegerita*

*Cyclocybe aegerita* (*Agrocybe aegerita*), also known as pioppino mushroom, is a basidiomycete belonging to the Strophariaceae family. Eleven STSs have been identified, all of which are heterologously expressed in *E. coli* [38]. Two of these (*Agr10* and *Agr11*) failed to detect the product. Using NPP as the precursor, δ-cadinene (**6**) was synthesized from *Agr1* and *Agr4*, and viridiflorene (**25**) was synthesized from *Agr2* and *Agr5* by C1,10 cyclization. Using FPP as a precursor, *Agr3* synthesized α-muurolene (**5**) by C1,10 cyclization, *Agr6* and *Agr7* synthesized Δ^6^-protoilludene (**11**), and *Agr8* synthesized γ-muurolene (**26**) by C1,11 cyclization. *Agr9* synthesized an unknown sesquiterpene alcohol.

#### 3.1.8. *Armillaria gallica*

*Armillaria gallica* is a saprophytic or parasitic fungus belonging to the Physalacriaceae family. Using bioinformatics analysis of its genome, 20 STSs were predicted [38], but only 1 (*Pro1*) was heterologously expressed in *E. coli* [39]. Using FPP as a precursor, *Pro1* synthesized Δ^6^-protoilludene (**11**) via C1,11 cyclization.

#### 3.1.9. *Galerina marginata*

*Galerina marginata* is a common poisonous mushroom belonging to the Hymenogastraceae family that contains amino peptides. Only one related STS (*Galma_104215*) has been identified and is heterologously expressed in *E. coli* [38]. Using NPP as the precursor, *Galma_104215* synthesized β-gurjunene (**27**) by C1,10 cyclization.

#### 3.1.10. *Omphalotus olearius*

*Omphalotus olearius* belongs to the family Omphalotaceae and emits green fluorescence. Ten STSs from this fungus have been identified [30]. *E. coli* was used for heterologous expression. With FPP as the precursor, *Omp1* and *Omp3* were used to synthesize α-muurolene (**5**) by C1,10 cyclization, and *Omp5a/b* was used to synthesize γ-cadinene (**3**); *Omp6* and *Omp7* synthesized Δ^6^-protoilludene (**11**) by C1,11 cyclization. With NPP as the precursor, *Omp4* was used to synthesize δ-cadinene (**6**) by C1,10 cyclization, *Omp9* synthesized α-barbatene (**28**), and *Omp10* mainly guided the synthesis of (*E*)-dauca-4(11), 8-diene (**29**) by C1,6 cyclization. *Omp8* is a homologue of *Omp9/10* that lacks approximately 100 amino acids at its N terminus and was not functional when expressed in *E. coli*.

### 3.2. Polyporales

Polyporales contains approximately 1800 species of fungi, representing approximately 1.5% of all known fungal species [40]. At present, 8 species of fungi in this order have been identified to contain 39 different STSs and 1 fusion enzyme.

#### 3.2.1. *Lignosus rhinocerus*

*Lignosus rhinocerus* (*Lignosus rhinocerotis*), also known as tiger milk mushroom, is a macrofungal belonging to the Polyporaceae family. Twelve terpene synthase genes have been found [41], seven of which are actively expressed in the sclerotium. Three STSs (*GME3634*, *GME3638*, and *GME9210*) were heterologously expressed by *S. cerevisiae*, producing nineteen, eight, and two sesquiterpenes, respectively (29 in total). Using FPP as the precursor, through C1,10 cyclization, *GME3634* mainly synthesized α-cadinol (**30**), and *GME3638* mainly synthesized torreyol (**31**). Using FPP as the precursor, *GME9210* mainly synthesized 1,3,4,5,6,7-hexahydro-2,5,5-trimethyl-2H-2,4a-ethanonaphthalene (**32**) and 1-napthalenol (**33**).

#### 3.2.2. *Cerrena unicolor*

*Cerrena unicolor* belongs to the Polyporaceae family. A total of 14 STSs have been found. Heterologous expression in *E. coli* was performed [42]. Four of these (*Cun5765*, *Cun6114*, *Cun7487*, and *Cun0802*) were not produced, and the loss of *Cun6114*, *Cun7487*, and *Cun5765* activities might be caused by the difficulty in predicting introns [43]. The failure of *Cun0802* gene cloning may be related to its low transcription level [44,45], and the product of *Cun9106* could not be identified. The other 9 STSs produced 10 different sesquiterpenes. Using NPP as the precursor, β-cubebene (**34**) was synthesized by *Cun3157*, and δ-cadinene (**6**) was synthesized by *Cun3158*. Using FPP as a precursor, δ-cadinol (**20**) was synthesized by *Cun7050*, α-copaene (**35**) was synthesized by *Cun3574*, α-muurolene (**5**) was synthesized by *Cun0759*, γ-cadinene (**3**) was synthesized by *Cun3817*, and germacrene D (**36**) was synthesized by *Cun0773*. Aromadendrene (**37**) was synthesized by C1,11 cyclization of *Cun5155* with FPP as the precursor; δ-cadinol (**20**) was synthesized by C1,6 cyclization from *Cun0716* with NPP as the precursor.

#### 3.2.3. *Rhodonia placenta*

*Rhodonia placenta* (*Postia placenta*), formerly known as brown rot fungus, is a common diseased wood-rot fungus belonging to the Polyporaceae family that can grow a large area of mycelium. It is known that 6 STSs have been isolated from it; they were heterologously expressed by *S. cerevisiae.* A total of 25 different sesquiterpenoids were synthesized with FPP or NPP precursors [46]. Using FPP as a precursor, through C1,10 cyclization, *PpSTS01* successfully synthesized α-muurolene (**5**), δ-cadinene (**6**), and β-elemene (**21**), *PpSTS03* synthesized α-cadinene (**38**) and γ-cadinene (**3**), and *PpSTS06* synthesized α-gurjunene (**39**); Δ^6^-protoilludene (**11**) and pentalenene (**13**) were synthesized from *PpSTS08* and *PpSTS14*, respectively, by C1,11 cyclization. δ-Cadinene (**6**) was synthesized from *PpSTS10* by C1,10 cyclization.

#### 3.2.4. *Fomitopsis pinicola*

*Fomitopsis pinicola* is a brown rot basidiomycete species belonging to the family Fomitopsidaceae, commonly collected from dead conifer trees. One STS (*Fompi1*) was identified and heterologously expressed in *E. coli* [30]. α-Cuprenene (**7**) was synthesized through C1,6 cyclization by *Fompi1* with NPP as the precursor.

#### 3.2.5. *Ganoderma lucidum*

*Ganoderma lucidum*, which belongs to the Ganodermataceae family, is a well-known medicinal fungus. However, only two STSs, *GL26009* [47] and *GISTS6* [48], have been isolated and identified from this fungus and expressed heterologously in *E. coli*. Using FPP as the precursor, *GL26009* synthesized γ-muurolene (**26**) and α-muurolene (**5**), and *GISTS6* synthesized γ-cadinene (**3**).

#### 3.2.6. *Ganoderma sinense*

*Ganoderma sinense* is a medicinal fungus belonging to the same genus as *G. lucidum* in the Ganodermataceae family. At present, six STSs have been isolated and identified from this fungus and expressed in *E. coli*. (*GS11330*, *GS14272*, *GS02363*, *GsSTS43*, *GsSTS45a*, and *GsSTS45b*) [48,49,50]. *GsSTS45a* has no function; *GS02363* synthesized α-cadinol (**30**), δ-cadinene (**6**), α-muurolene (**5**), and γ-muurolene (**26**); *GS11330* synthesized α-cuprenene (**7**); *GS14272* synthesized α-muurolene (**5**); and *GsSTS43* and *GsSTS45b* synthesized γ-cadinene (**3**).

#### 3.2.7. *Phanerodontia chrysosporium*

*Phanerodontia chrysosporium* (*Phanerochaete chrysosporium*) belongs to the family Phanerochaetaceae. Eleven STSs have been recorded [51], of which seven were heterologously expressed in *S. cerevisiae* and cultured in SDL medium. *PcSTS01* synthesized γ-muurolene (**26**), α-muurolene (**5**), and δ-cadinene (**6**). *PcSTS02* and *PcSTS04* synthesized β-copaene (**40**),β-farnesene (**22**), cadina-1(**6**),4-diene (**41**), and δ-cadinene (**6**). Epicubenol (**42**) was synthesized from *PcSTS03*. *PcSTS06* synthesized α-barbatene (**28**) and β-barbatene (**43**). (*E*)-α-Bisabolene (**44**) was synthesized from *PcSTS08*, and *PcSTS11* synthesized α-santalene (**45**).

#### 3.2.8. *Steccherinum ochraceum*

*Steccherinum ochraceum* belongs to the family Meruliaceae. Six STSs were deduced from its genome, of which fusion enzyme *A8411* was heterologously expressed in *A. oryzae* [52]. Hirsutene (**46**) was synthesized from *A8411*.

### 3.3. Russulales

Russulales comprises approximately 1767 species belonging to 80 genera and 12 families [1]. Two fungi of this order have been verified to contain fifteen different STSs and one fusion enzyme.

#### 3.3.1. *Stereum hirsutum*

*Stereum hirsutum* belongs to the family Stereaceae. Nearly 50 related sesquiterpenoids have been found [53] as well as 18 STSs (*ShSTS1-18*, of which *ShSTS2*, *6*, *9*, *14* have not been studied due to their high homology to other genes) and 1 fusion protein (*HS-HMGS*), which were functionally verified by heterologous expression in *E. coli* or *A. oryzae* [36,38,54,55]. Their synthetic routes are summarized as follows: using NPP as a substrate, via C1,10 cyclization, *ShSTS10* and *ShSTS11* can synthesize δ-cadinene (**6**), *ShSTS8* can synthesize 1-epi-cubenol (**47**), *ShSTS12* can synthesize α-cubebene (**48**), and *ShSTS10* can synthesize germacrene D (**36**). Simultaneously, *ShSTS1* synthesized β-barbatene (**43**), *ShSTS3* synthesized α-farnesene (**14**) and β-farnesene (**22**), *ShSTS4* synthesized hirsutene (**46**), and *ShSTS5* synthesized γ-cadinene (**3**) by C1,6 cyclization. Using FPP as a substrate, through C1,11 cyclization, *ShSTS13* can synthesize β-caryophyllene (**23**), *HS-HMGS* can synthesize hirsutene (**46**), and *ShSTS15*, *ShSTS16*, *ShSTS17*, and *ShSTS18* can synthesize Δ^6^-protoilludene (**11**); *ShSTS7* synthesized δ-cadinene (**6**) via C1,10 cyclization.

#### 3.3.2. *Heterobasidion annosum*

*Heterobasidion annosum* belongs to the Bondarzewiaceae family of the order Russulales. It is a forest pathogen that grows on large, perennial basidiocarps. Only one STS (*Hetan2_454193*) was identified in this fungus [40] and was heterologously expressed in *E. coli*. Using FPP as the precursor, *Hetan2_454193* synthesized Δ^6^-protoilludene (**11**) by C1,11 cyclization.

### 3.4. Other Basidiomycota

The basidiomycetes in this region cannot be classified by order. There are 6 species of fungi in this part, and 10 different STSs have been verified by experiments.

#### 3.4.1. *Boreostereum vibrans*

*Boreostereum vibrans*, originally named *Stereum vibrans*, is a macrofungus belonging to the Gloeophyllaceae family of the order Gloeophyllales. Many sesquiterpenes have been isolated from it [56]. *BvCS* is heterologously expression in *E. coli* [57]. δ-Cadinol (**20**) was synthesized by C1,10 cyclization of *BvCS* with FPP as a precursor.

#### 3.4.2. *Sphaerobolus stellatus*

*Sphaerobolus stellatus* belongs to the order Geastrales and family Geastraceae. One STS (*Sphst_47084*) has been identified in this fungus [38] and heterologously expressed in *E. coli*. Using NPP as the precursor, viridiflorol (**49**) was synthesized via C1,10 cyclization of *Sphst_47084*.

#### 3.4.3. *Sanghuangporus baumii*

*Sanghuangporus baumii* belongs to the Hymenochaetaceae family of the order Hymenochaetales and is an important medicinal fungus. Only one STS has been isolated from this species and heterologously expressed by *E. coli*, named *SbTps1* [58].

#### 3.4.4. *Coniophora puteana*

*Coniophora puteana* belongs to the Coniophoraceae family within the order Boletales. Four STSs have been isolated from it (*Copu2*, *3*, *5* and *9*) [59,60]. β-Copaene (**40**) and cubebol (**9**) were synthesized by C1,10 cyclization from *Copu2* and *Copu3* with NPP as the precursor. Using FPP as the precursor, *Copu5* and *Copu9* synthesized δ-cadinol (**20**) through C1,10 cyclization.

#### 3.4.5. *Serendipita indica*

*Serendipita indica* is an endophytic root-colonizing species belonging to the order Sebacinales and family Serendipitaceae. One STS has been recorded [61], which is hetero-expressed in *E. coli*. Viridiflorol (**49**) was synthesized by C1,10 cyclization from *SiTPS*, using FPP as the precursor.

#### 3.4.6. *Dendrodontia bispora*

*Dendrodontia bispora* (*Dendrothele bispora*) is a basidiomycete species belonging to the Corticiaceae family in Corticiales order. Two STSs (*Denbi1_659367* and *Denbi1_816208*) have been isolated from this fungus [38] and hetero-expressed in *E. coli*. Δ^6^-protoilludene (**11**) was synthesized by C1,11 cyclization of *Denbi1_659367*. Viridiflorol (**49**) was synthesized by *Denbi1_816208.*

The taxonomic data in the above content come from GBIF (Global Biodiversity Information Facility, https://www.gbif.org/, accessed on 28 June 2022). A summary of information on sesquiterpene biosynthesis in Basidiomycota is presented in Table 1.

## 4. Research Process and Tools for STSs in Basidiomycetes

At present, the research process of basidiomycete STSs is mainly divided into three parts (Figure 4), and the latest research tools are developed around the core steps of these three parts (genome sequencing, basidiomycete culture methods, etc.).

### 4.1. Long-Read Whole-Genome Sequencing

The average genome size of basidiomycetes and ascomycetes is 46 Mb and 37 Mb, respectively [62], and the genome size of archaea and bacteria is usually within 6 Mb [63]. This means that bacterial genomes can be sequenced using short-read sequencing, but long-read sequencing and proper assembly are required for fungal genomes. Inexpensive nanopore sequencing [64] has been applied to whole-genome sequencing of basidiomycetes. For example, nanopore sequencing technology was used for whole-genome sequencing of the basidiomycete *Clathrus columnatus* and *Inonotus obliquus*, and genome assembly was completed [65,66].

Although nanopore technology enables long-read sequencing and is inexpensive, the accuracy of base calling is 85–94% depending on the sequencing method [67]. However, if nanopore long-read sequencing is used in combination with short-read sequencing, it may be possible to assemble a higher quality genome, if the software that assembles the genome has this capability. This functionality is currently available for bacterial and fungi genomes, and the Pilon software can combine Illumina and Nanopore sequence data to polish assemblies [68].

### 4.2. Basidiomycetes Cultures

Many basidiomycetes have high requirements for their growth environment; they can only grow under specific conditions, and most of them cannot be cultivated artificially, which greatly limits isolation and identification of sesquiterpenoids in basidiomycetes. Fungal growth can be simply divided into two stages: (i) germination of fungal spores, and (ii) subsequent filamentous growth, forming a network of hyphae called mycelium [69].

The conditions that induce or inhibit the germination of fungal spores have always puzzled researchers. So far, the main research directions are growth factor induction, activator induction, co-culture, volatile organic compound induction, and physical factors [70]. Taking Agaricomycetes ectomycorrhizal (EcM) mushrooms as an example, the M factor (growth-promoting metabolites in addition to b vitamins and amino acids are essential for the growth of tree mycorrhizal fungi), as a growth factor, promotes its growth [71]. Placing EcM mushrooms together with specific tree seedlings on lipid or gel medium can also promote spore germination, although this approach often fails [71,72]. There is evidence that EcM mushroom spore germination can also be promoted when co-cultured with bacteria [73].

The growth factors that induce the growth of fungal hyphae are mostly root exudates. Studies on EcM fungi have shown that in addition to M factor, palmitic acid, stearic acid, and cytokinins, such as kinetin, zeatin, and isopentenyl aminopurine, are also growth-promoting factors. Root exudates can induce their growth [74].

At the same time, fungal gene expression is very complex and is affected by RNAi silencing [75] and trans-acting elements of genome structure [76]. Therefore, many biosynthetic gene clusters are silent. However, by adjusting the culture conditions (changing the physical conditions of the culture, adding compounds, growth factors, etc.) [70], utilizing co-culture [77], and chromatin-based transcriptional regulation [78], silenced biosynthetic gene clusters can be activated. Studies on these operations are still in preliminary stages. However, the metabolites produced by basidiomycete fungi in different growth cycles are different [79], and many genetic regulators that control fungal development also control the production of secondary metabolites [80,81]. Studies on the basidiomycetes *Coprinopsis cinerea* [82] and *Lentinula edodes* [83] have shown that gene expression differs at the developmental stages of fruiting bodies, limiting the mining of active ingredients.

Cultivation technology for basidiomycetes has always been inadequate, but in recent years, the development of new laboratory-level cultivation techniques has brought new opportunities for artificial cultivation. For example, basidiomycetes are cultivated using microfluidic culture technology [84].

### 4.3. Exogenous Expression Platforms and Bioinformatics Tools for Basidiomycetes STSs

Due to the complexity of basidiomycetes genes, traditional heterologous expression platforms cannot meet the functional identification of basidiomycetes STS. Although *E. coli* can express STS genes, it lacks a post-translational modification system to express complex proteins and entire biosynthetic pathways, and the eukaryotic expression system in yeast cannot remove introns of fungal genes [85]. At present, *A. oryzae* is a relatively successful heterologous expression platform, which can more accurately splice the intron of the basidiomycetes terpenoid synthase gene [37] and correctly express the entire gene cluster [86]. *Ustilago maydis* has also been developed as a heterologous expression platform for the production of terpenoids [87]. It offers the advantage of metabolic compatibility and potential tolerance of substances toxic to other microorganisms.

Successful characterization of the biosynthesis of basidiomycetes products requires not only genetic engineering and heterologous expression, but also metabolic analysis [88]. Bioactivity-guided methods for isolating metabolites have been gradually replaced by more sensitive methods, such as tandem mass spectrometry (MS/MS), for untargeted metabolomics data analysis, resulting in data that can be compared with known spectral databases. Researchers can also identify unknown metabolites and intermediates through the Global Natural Products Social Molecular Networking (GNPS) [88,89] and infer biosynthetic pathways. New technologies and tools are also being developed to assist in the identification of STSs in basidiomycetes. Bioinformatics techniques can be used to establish a general prediction framework for STS and to improve the accuracy of genome-based tools for predicting biosynthetic gene clusters [58], such as antiSMASH [90] and PRISM [91]. Although it can predict the monomer sequences assembled into PKS and NRPS biosynthetic lines based on module specificity, the accuracy and specificity need to be further improved, which is also the key to identifying STS [91].

## 5. Discussion

Sesquiterpenes play a very important role in basidiomycetes. They can attract insects for pollination [92], defend against other organisms or parasites [93], and play an important role in basidiomycete’s physiological effect. Moreover, when basidiomycetes form a symbiotic relationship with plants, these sesquiterpenoids produced by basidiomycetes can act as phytohormones [13,14]. Therefore, basidiomycetes produce many kinds of sesquiterpenes to help them better adapt to the living environment. The influence of the ambient environment on the production of sesquiterpenes by basidiomycetes includes external physical factors (light, temperature, etc.) and chemical factors (exogenous chemical substances, etc.). The changes of sesquiterpenes during basidiomycete development and their biological roles are still unclear. At present, there are studies on the changes of sesquiterpenes during the development of the fruiting bodies of the *Cyclocybe aegerita* AAE-3 strain. In particular, the development of the fruiting body changes resulted in greater changes during the sporulation process. In the early stage of sporulation, mainly alcohols and ketones appeared, while in the later stage of sporulation, sesquiterpenes such as Δ6-protoilludene (**11**), α-cubebene (**48**) and δ-cadinene (**6**) appeared. After sporulation, sesquiterpenoids decreased and other compounds appeared, mainly octan-3-one [94].

Site-specific mutations are tools to study enzyme structure, function, and catalytic mechanism, and they include single and combinatorial mutations [95]. In the study of sesquiterpene synthases of basidiomycetes, the point mutations at residues near the conserved region are mostly used. The current point mutation experiments for sesquiterpene synthase of basidiomycetes are concentrated near the conserved regions of the RY Pair and NSE Triad (Figure 5).

*Cop3*, *Cop4*, and *Cop6* were experimentally point mutated at the sites of their H-α1 loops, respectively. K233 in *Cop4* and K251 in *Cop3* did not play a major role in the side chain and ligand interaction network formed during active-site closure; the mutations in *Cop4* (K233, H235, T236, N238, and N239) showed that the mutations in the H-α1 loop region site significantly altered the type of product, with the mutation of N239L having the greatest effect on the product; the mutation of *Cop6* (C236, E237, and N240) showed that the mutation of the H-α1 loop region site did not alter the product of *Cop6*. Structural modeling of the Cop enzyme pointed to a potential interaction between the H-α1 loop and the conserved residues of the two metal-binding motifs (DDXXDD and NSE/DTE). Potential interactions between the conserved Asp/Glu and the Arg, Asn and Lys sites in some sesquiterpene synthases in several fungi and plants may stabilize the closed enzyme conformation by closing the H-α1 loop [96]. *STC4* was transformed into germacrene A (**4**) synthase after the single site W335F mutation; various variants of the W314 point mutation in *STC15* were unable to obtain expressed protein, and enzyme activity was reduced after the mutation of the putative C311 active site in *STC9* [34]. A triple mutant *Cop2*(17H2) was obtained by error-prone PCR. *Cop2*(17H2) contains three mutations in L59H, T65A and S310Y, and the three mutations tend to make *Cop2*(17H2) products be specific. Moreover, compared to the original *Cop2*, *Cop2*(17H2) is more inclined to produce Germacrene D-4-ol (**2**) [97].

## 6. Conclusions

A comparison of the reported genome sequences revealed that each basidiomycete contained, on average, more than 12 STSs. Although the reasons for the existence of many STSs are unclear, it is speculated that they are closely related to their biological activities. The development of molecular tools for basidiomycetes research will allow researchers to further explore these microbial taxa. These efforts have definitely resulted in a global push for the discovery and characterization of fungal STSs, and they provide hope for the future of fungal sesquiterpenoid discovery.

## Figures and Tables

**Figure 1 jof-08-00913-f001:**
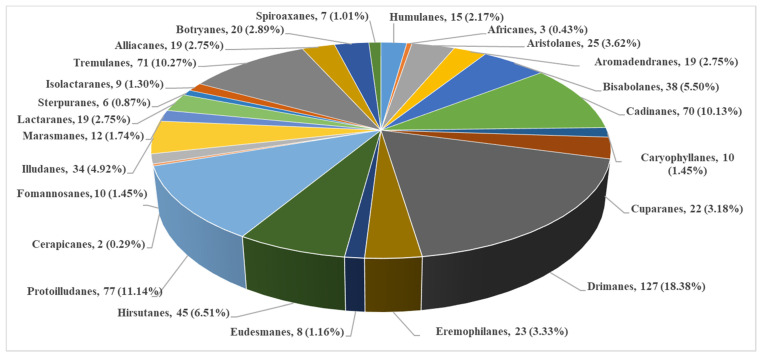
Basidiomycota sesquiterpenes classified by a skeleton.

**Figure 2 jof-08-00913-f002:**
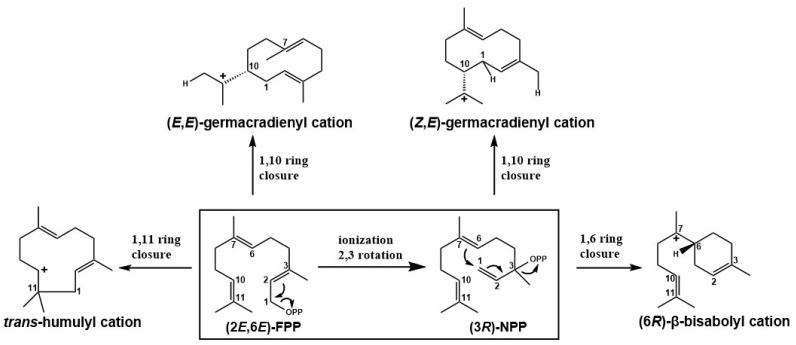
Cyclization patterns of sesquiterpenes in basidiomycetes.

**Figure 3 jof-08-00913-f003:**
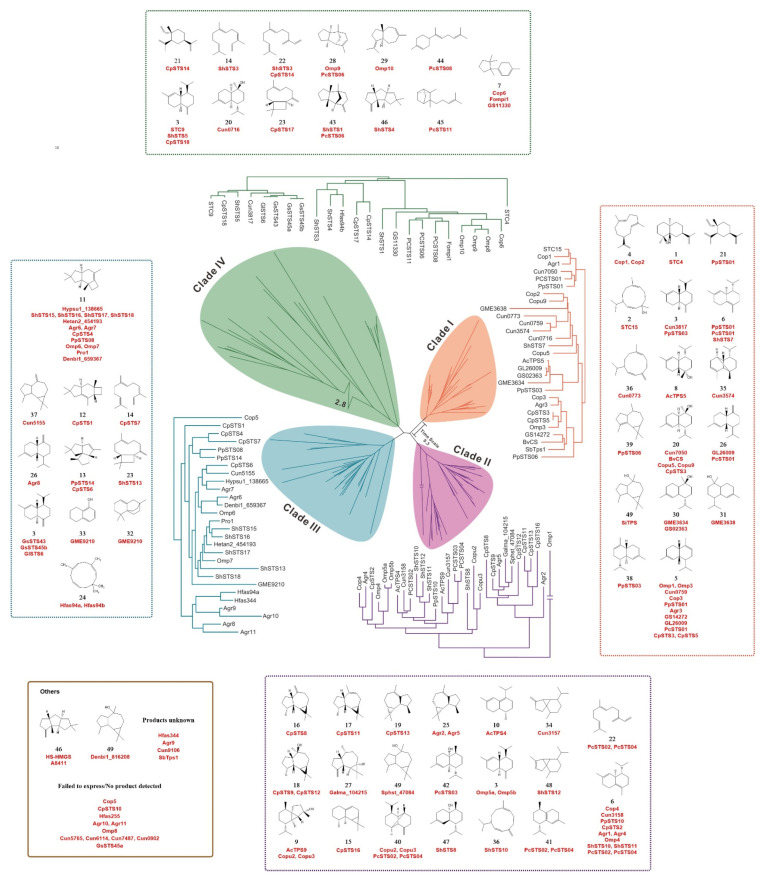
Sesquiterpene biosynthesis by STS of clades I–IV in basidiomycetes. All mentioned STSs have undergone biochemical verification.

**Figure 4 jof-08-00913-f004:**
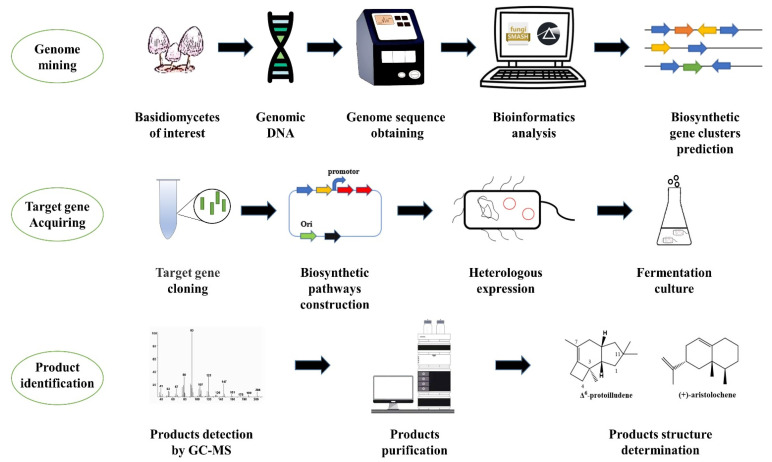
Technical research route of STS in basidiomycetes. The STS genome mining of basidiomycetes can be divided into three steps. The first step of genome mining is mainly based on the whole gene sequence, using bioinformatics tools to mine and predict the biosynthetic gene cluster. The second step is to obtain the target gene from the biosynthetic gene cluster of STS, and then heterologously express the target gene to obtain the product. The final step is to purify and identify the product to determine whether the gene is an STS.

**Figure 5 jof-08-00913-f005:**
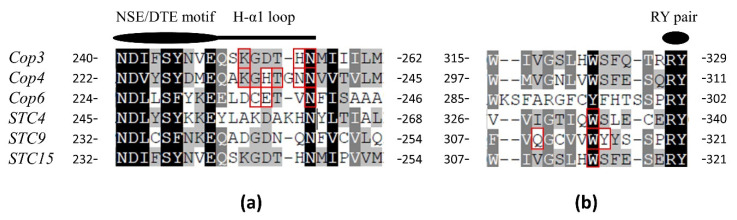
(**a**) Comparison of the conserved regions of H-α1 loop and NSE/DTE motif. (**b**) Comparison of conserved regions of the RY Pair to the Thirteen Positions Upstream of the RY Pair. The MUSCLE algorithm was used to compare the following protein sequences: *Cop3* (XP_001832925), *Cop4* (XP_001836356), *Cop6* (XP_001832549) in *Coprinopsis cinerea*; *STC4* (KAH0582448), *STC9* (KAH0583476), *STC15* (KAG5341349) in *Macrolepiota albuminosa*.

**Table 1 jof-08-00913-t001:** Classification of STSs from Basidiomycota.

	Type of Cyclization	Precursor	Metabolite	Producer	Biochemically Verified Enzyme
Clade I	C1,10	FPP	Intermedeol (**1**)	*Macrolepiota albuminosa*	*STC4*
			Germacrene D-4-ol (**2**)	*Macrolepiota albumi-nosa*	*STC15*
			Germacrene A (**4**)	*Coprinopsis cinerea*	*Cop1, Cop2*
			T-Cadinol (**8**)	*Taiwanofungus camphoratus*	*AcTPS5*
			α-Muurolene (**5**)	*Omphalotus olearius*	*Omp1, Omp3*
				*Cerrena unicolor*	*Cun0759*
				*Coprinopsis cinerea*	*Cop3*
				*Rhodonia placenta*	*PpSTS01*
				*Cyclocybe aegerita*	*Agr3*
				*Ganoderma sinense*	*GS14272*
				*Ganoderma lucidum*	*GL26009*
				*Phanerodontia chrysosporium*	*PcSTS01*
				*Clitopilus pseudo-pinsitus*	*CpSTS3, CpSTS5*
			α-Cadinol (**30**)	*Lignosus rhinoceros* *Ganoderma sinense*	*GME3634* *GS02363*
			Torreyol (**31**)	*Lignosus rhinocerus*	*GME3638*
			δ-Cadinol (**20**)	*Cerrena unicolor*	*Cun7050*
				*Boreostereum vibrans*	*BvCS*
				*Coniophora puteana*	*Copu5, Copu9*
				*Clitopilus pseudo-pinsitus*	*CpSTS3*
			α-Copaene (**35**)	*Cerrena unicolor*	*Cun3574*
			γ-Cadinene (**3**)	*Cerrena unicolor*	*Cun3817*
				*Rhodonia placenta*	*PpSTS03*
			Germacrene D (**36**)	*Cerrena unicolor*	*Cun0773*
			δ-Cadinene (**6**)	*Rhodonia placenta*	*PpSTS01*
				*Phanerodontia chrysosporium*	*PcSTS01*
				*Stereum hirsutum*	*ShSTS7*
			β-Elemene (**21**)	*Rhodonia placenta*	*PpSTS01*
			α-Cadinene (**38**)	*Rhodonia placenta*	*PpSTS03*
			α-Gurjunene (**39**) Viridiflorol (**49**) γ-Muurolene (**26**)	*Rhodonia placenta* *Serendipita indica* *Ganoderma lucidum* *Phanerodontia chrysosporium*	*PpSTS06* *SiTPS* *GL26009* *PcSTS01*
Clade II	C1,10	NPP	δ-Cadinene (**6**)	*Coprinopsis cinerea*	*Cop4*
				*Cerrena unicolor*	*Cun3158*
				*Rhodonia placenta*	*PpSTS10*
				*Clitopilus pseudo-pinsitus*	*CpSTS2*
				*Cyclocybe aegerita*	*Agr1, Agr4*
				*Omphalotus olearius*	*Omp4*
				*Stereum hirsutum* *Phanerodontia chrysosporium*	*ShSTS10, ShSTS11* *PcSTS02, PcSTS04*
			Cubebol (**9**)	*Taiwanofungus camphoratus* *Coniophora puteana*	*AcTPS9* *Copu2, Copu3*
			1-epi-Cubenol (**47**)	*Stereum hirsutum*	*ShSTS8*
			Zonarene (**10**)	*Taiwanofungus camphoratus*	*AcTPS4*
			Ledene (**19**)	*Clitopilus pseudo-pinsitus*	*CpSTS13*
			Virifloridol (**18**)	*Clitopilus pseudo-pinsitus*	*CpSTS9, CpSTS12*
			Viridiflorol (**49**)	*Sphaerobolus stellatus*	*Sphst_47084*
			Alloaromadendrene (**16**)	*Clitopilus pseudo-pinsitus*	*CpSTS8*
			9-Alloaromadendrene (**17**)	*Clitopilus pseudo-pinsitus*	*CpSTS11*
			Aristolene (**15**)	*Clitopilus pseudo-pinsitus*	*CpSTS16*
			Viridiflorene (**25**)	*Cyclocybe aegerita*	*Agr2, Agr5*
			γ-Cadinene (**3**)	*Omphalotus olearius*	*Omp5a, Omp5b*
			α-Cubebene (**48**)	*Stereum hirsutum*	*ShSTS12*
			β-Cubebene (**34**)	*Cerrena unicolor*	*Cun3157*
			Germacrene D (**36**)	*Stereum hirsutum*	*ShSTS10*
			β-Copaene (**40**)	*Coniophora puteana* *Phanerodontia chrysosporium*	*Copu2, Copu3* *PcSTS02, PcSTS04*
			β-Gurjunene (**27**) Epicubenol (**42**) β-Farnesene (**22**) Cadina-1(6),4-diene (**41**)	*Galerina marginata* *Phanerodontia chrysosporium* *Phanerodontia chrysosporium* *Phanerodontia chrysosporium*	*Galma_104215* *PcSTS03* *PcSTS02, PcSTS04* *PcSTS02, PcSTS04*
Clade III	C1,11	FPP	α-Humulene (**24**)	*Hypholoma fasciculare*	*Hfas94a, Hfas94b*
			Δ^6^-Protoilludene (**11**)	*Hypholoma lateritium*	*Hypsu1_138665*
				*Stereum hirsutum*	*ShSTS15, ShSTS16, ShSTS17, ShSTS18*
				*Heterobasidion annosum*	*Hetan2_454193*
				*Cyclocybe aegerita*	*Agr6, Agr7*
				*Clitopilus pseudo-pinsitus*	*CpSTS4*
				*Rhodonia placenta*	*PpSTS08*
				*Omphalotus olearius*	*Omp6, Omp7*
				*Armillaria gallica*	*Pro1*
				*Dendrodontia bispora*	*Denbi1_659367*
			Aromadendrene (**37**)	*Cerrena unicolor*	*Cun5155*
			β-Caryophyllene (**23**)	*Stereum hirsutum*	*ShSTS13*
			Pentalenene (**13**)	*Rhodonia placenta*	*PpSTS14*
				*Clitopilus pseudo-pinsitus*	*CpSTS6*
			Sterpurene (**12**)	*Clitopilus pseudo-pinsitus*	*CpSTS1*
			α-Farnesene (**14**)	*Clitopilus pseudo-pinsitus*	*CpSTS7*
			γ-Muurolene (**26**)	*Cyclocybe aegerita*	*Agr8*
			1,3,4,5,6,7-Hexahydro-2,5,5-trimethyl-2H-2,4a-ethanonaphthalene (**32**)	*Lignosus rhinocerus*	*GME9210*
			1-Napthalenol (**33**)	*Lignosus rhinocerus*	*GME9210*
			γ-Cadinene (**3**)	*Ganoderma sinense*	*GsSTS43, GsSTS45b*
				*Ganoderma lucidum*	*GISTS6*
Clade IV	C1,6	NPP	α-Cuprenene (**7**)	*Coprinopsis cinerea* *Ganoderma sinense* *Fomitopsis pinicola*	*Cop6* *GS11330* *Fompi1*
			α-Barbatene (**28**)	*Omphalotus olearius* *Phanerodontia chrysosporium*	*Omp9* *PcSTS06*
			β-Barbatene (**43**)	*Phanerodontia chrysosporium*	*PcSTS06*
				*Stereum hirsutum*	*ShSTS1*
			α-Farnesene (**14**)	*Stereum hirsutum*	*ShSTS3*
			β-Farnesene (**22**)	*Stereum hirsutum* *Clitopilus pseudo-pinsitus*	*ShSTS3* *CpSTS14*
			Hirsutene (**46**)	*Stereum hirsutum*	*ShSTS4*
			γ-Cadinene (**3**)	*Termitomyces* *albuminosus*	*STC9*
				*Stereum hirsutum*	*ShSTS5*
				*Clitopilus pseudo-pinsitus*	*CpSTS18*
			β-Elemene (**21**)	*Clitopilus pseudo-pinsitus*	*CpSTS14*
			β-Caryophyllene (**23**)	*Clitopilus pseudo-pinsitus*	*CpSTS17*
			(*E*)-Dauca-4(**11**),8-diene (**29**)	*Omphalotus olearius*	*Omp10*
			δ-Cadinol (**20**)	*Cerrena unicolor*	*Cun0716*
			(*E*)-α-Bisabolene (**44**)	*Phanerodontia chrysosporium*	*PcSTS08*
			α-Santalene (**45**)	*Phanerodontia chrysosporium*	*PcSTS11*
others			—	*Coprinopsis cinerea*	*Cop5*
			—	*Clitopilus pseudo-pinsitus*	*CpSTS10*
			—	*Hypholoma fasciculare*	*Hfas255*
			Unknown	*Hypholoma fasciculare*	*Hfas344*
			Unknown	*Cyclocybe aegerita*	*Agr9*
			—	*Cyclocybe aegerita*	*Agr10, Agr11*
			—	*Omphalotus olearius*	*Omp8*
			—	*Cerrena unicolor*	*Cun5765, Cun6114, Cun7487, Cun0802*
			Unable to identify	*Cerrena unicolor*	*Cun9106*
			—	*Ganoderma sinense*	*GsSTS45a*
			Unknown	*Sanghuangporus baumii*	*SbTps1*
			Viridiflorol (**49**)	*Dendrodontia bispora*	*Denbi1_816208*
			Hirsutene (**46**)	*Stereum hirsutum*	*HS-HMGS*
				*Steccherinum ochraceum*	*A8411*

## Data Availability

Not applicable.

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
