# Peer review of "Biosynthesis of Sesquiterpenes in Basidiomycetes: A Review"

_jof, 2022, doi:10.3390/jof8090913_

Round 1
Reviewer 1 Report
This review summarizes sesquiterpene synthase identified from different species of basidiomycetes over the past 20 years. The biological functions of these enzymes and related compound structures are described. Additionally, research process and tools for sesquiterpene synthase in basidiomycete are discussed. In general, the manuscript is well organized, which can be published with some minor revision as following:
1. Please carefully check the usage of “Basidiomycetes” and “basidiomycetes” in the manuscript.
2. Line 56-57: sesquiterpene genes or gene clusters?
3. Line 71-72:” The second step involves FPP generating different sesquiterpene 71 carbon skeletons through irregular coupling reactions.” The sentence is hard to read.
4. Line 114: “STC4 was cyclized by C1,10 cyclization to synthesize intermedeol” The sentence is misleading.
5. Line 115-116: The expression is not clear. STC9 can synthesis both compound 2 and 3?
6. Line 127-128: It’s not clear whether 10 terpene synthases were all heterologously expressed in E.coli ?
7. Line 136: It seems should be remain 11 STSs.
8. Line 268-271: The description is not appropriate, “HS-HMGS” should not be include in STS.
9. Line 349: It’s better to check whether the software for fungi is already available?
10. It would be better to briefly describe the difference between A. oryzae and Ustilago maydis expression host.
Author Response
Reviewer #1:
This review summarizes sesquiterpene synthase identified from different species of basidiomycetes over the past 20 years. The biological functions of these enzymes and related compound structures are described. Additionally, research process and tools for sesquiterpene synthase in basidiomycete are discussed. In general, the manuscript is well organized, which can be published with some minor revision as following:
Response: Thank you very much for your comments. For your questions involved in the manuscript, we have seriously revised them.
- Please carefully check the usage of “Basidiomycetes” and “basidiomycetes” in the manuscript.
Response: As suggested, we carefully checked the usage of “Basidiomycetes” and “basidiomycetes” in the manuscript.
- Line 56-57: sesquiterpene genes or gene clusters?
Response: Thanks for your valuable feedback, "sesquiterpene genes" should be used here after checking.
- Line 71-72: “The second step involves FPP generating different sesquiterpene 71 carbon skeletons through irregular coupling reactions.” The sentence is hard to read.
Response: As suggested, we revised the sentence change to “As the second step, FPP generates different sesquiterpene carbon skeletons through irregular coupling reactions.”
- Line 114: “STC4 was cyclized by C1,10 cyclization to synthesize intermedeol” The sentence is misleading.
Response: Thanks for your valuable feedback, we have revised this sentence change to “STC4 synthesized intermedeol (1) via C1,10 cyclization,”
- Line 115-116: The expression is not clear. STC9 can synthesis both compound 2 and 3?
Response: Thanks for your valuable feedback, this sentence was missing a part in the previous modification, it has now been revised.
- Line 127-128: It’s not clear whether 10 terpene synthases were all heterologously expressed in E.coli ?
Response: Thanks for your valuable feedback, we have double checked and there is no problem here.
- Line 136: It seems should be remain 11 STSs.
Response: Thanks for your valuable feedback, we made a mistake there, have now double checked and added new content.
- Line 268-271: The description is not appropriate, “HS-HMGS” should not be include in STS.
Response: Thanks for your valuable feedback, we have made some revisions. The new description is “fusion protein (HS-HMGS)”
- Line 349: It’s better to check whether the software for fungi is already available?
Response: Thanks for your valuable feedback, we have made some revisions. This sentence is changed to “This functionality is currently available for bacterial genomes, and the Pilon software can combine Illumina and Nanopore sequence data to polish assemblies [69]. We believe that software for fungi will be available soon.”
- It would be better to briefly describe the difference between A. oryzae and Ustilago maydis expression host.
Response: As suggested, we added a new sentence “It offers the advantage of metabolic compatibility and potential tolerance of substances toxic to other microorganisms.”
Reviewer 2 Report
attached

Author Response
Beforehand: Such a review is missing in literature, and I appreciated this busy work, which should be published after thorough revision. I have received an already revised version, but it seems that the authors did not care much about the recommendations of the first reviewing round.
Response: Thank you very much for taking the time to comment on our article. Your comments were detailed and valuable. We read your comments carefully and made changes to the article and supporting materials in response to your suggestions.
A major problem is language, but also presentation in general, starting with the not very meaningful abstract over illegible Figures to incomplete references.
Response: We greatly appreciate the opinions and moral support from experts, the manuscript was edited in English language by Editage (www.editage.cn) once again.
Abstract: Sesquiterpenes are common small-molecule natural products with a wide range of promising applications, and their biosynthesis is regulated by sesquiterpene synthase (STS). Basidiomycetes are valuable and important biological resources. To date, hundreds of related sesquiterpenoids have been discovered in basidiomycetes, and the biosynthetic pathways of some of these compounds have been elucidated. This review summarizes 111 STSs identified from 25 species of basidiomycetes over the past 20 years. The biological functions of enzymes and compound structures are described, and related research is discussed.
My version:
Abstract: Sesquiterpenes are natural products with a wide range of applications, and their biosynthesis is catalyzed by sesquiterpene synthases (STS). Basidiomycetes are valuable biological resources of STS. To date, hundreds of related sesquiterpenoids have been discovered in basidiomycetes, and the biosynthetic pathways of some of these compounds have been elucidated. This review summarizes 111 STSs identified from 25 species of basidiomycetes over the past 20 years. The biological functions of enzymes and compound structures are described, and related research is discussed. More specific data on what is presented should follow here.
delete common small-molecule interested readers will know anyway, “common”
delete promising many applications exist in the flavour and fragrance field
replace regulated It is typically the genes that are regulated, not the STS delete andimportant No practical application of a basidiomycete source is known
Response: Thank you very much for your careful review. We have revised the abstract as you suggested.
Line 28 C15 are oligomers, “oligoisoprenoids”, more specifically trimers
L 29 sesquiterpenoid natural products sesquiterpenes from fungi are always natural
L 30 good biological activities “supposed” or “presumed” … there is little evidence
from human studies
L 50 reportedly obtained from “reported from” or “obtained from” L 72 containscontain (singular)
L 74 defocusing the catalytic reaction of phosphoric acid ? There is a diphosphate leaving
group, what is meant here?
I stop here and recommend to use MDPI´s editing services
Response: We sincerely apologize for the level of English presentation, the manuscript was edited in English language by Editage (www.editage.cn) once again. we have touched up our article, corrected incomplete or illogical sentences, and replaced some inappropriate words.
Figure 2 is almost illegible in the pdf
Response: As suggested, we have improved the clarity of the image and resized the text in the image to make it more legible.
Figure 3 is almost illegible, too, I recommend to separate the presentation of the four clades in four different Figures.
Response: Thanks for your valuable feedback, we did not split into four parts, the reader can view such a whole diagram, while compare the Table 1 we provide, can more easily get the information they use, and it easier to understand. However, we still made changes to this figure. We modified the clarity of the image and increased the font of the text in the figure to make the whole figure look and feel better.
Figure 4 zoom or enlarge lettering to allow easier reading
Response: Thanks for your valuable feedback, we enlarged the font of the text in the figure according to your suggestion.
Greek letter should be in italics throughout the paper.
Response: Thank you for your careful review, Greek letters were changed to italics in the manuscript
The file “jof-1777288-non-published 25.7.” shows that six out of the seven authors were engaged in “writing”. This is hard to believe, especially when looking at the numerous shortcomings. Name only the author(s), which has/have written, not the free riders.
Response: Thank you very much for your comments, there was a lack of consideration in the author's contribution, we have revised it in the new manuscript.
We promise there will be no free riding.
Five out of the seven were engaged in “Data Curation”. However, several recent references were overlooked. There might have been reasons for not including these papers, but then this reason should be given. Again, omit the free riders.
Examples:
Bioactive Sesquiterpenes from the Edible Mushroom Flammulina velutipes and Their Biosynthetic Pathway Confirmed by Genome Analysis and Chemical Evidence. Tao Q, Ma K, Yang Y, Wang K, Chen B, Huang Y, Han J, Bao L, Liu XB, Yang Z, Yin WB, Liu H. J Org Chem. 2016 21;81(20):9867-9877. doi: 10.1021/acs.joc.6b01971.
Response: This paper mapped the sesquiterpene synthase genome of Flammulina velutipes, but that did not experimentally validate individual genes; we wrote this review to include only the experimentally validated functional sesquiterpene synthases.
New terpenoids from the fermentation broth of the edible mushroom Cyclocybe aegerita. Surup, Frank ; Hennicke, Florian; Sella, Nadine; Stroot, Maria; Bernecker, Steffen; Pfuetze, Sebastian; Stadler, Marc ; Ruehl, Martin.Beilstein Journal of Organic Chemistry 2019, 15, 1000-1007
Response of the sesquiterpene synthesis in submerged cultures of the Basidiomycete Tyromyces floriformis to the medium composition. Grosse M, Strauss E, Krings U, Berger RG. Mycologia. 2019;111(6):885-894. doi: 10.1080/00275514.2019.1668740.
Bovistol B, bovistol D and strossmayerin: Sesquiterpene metabolites from the culture filtrate of the basidiomycete Coprinopsis strossmayeri. Banks AM, Song L, Challis GL, Bailey AM, Foster GD. PLoS One. 2020;15(4):e0229925. doi: 10.1371/journal.pone.0229925. eCollection 2020.
Response: These articles you cited are all studies only isolated sesquiterpenoids, without studying the function of biosynthetic genes, which we were unable to include in the main body of the article, and the characteristic sesquiterpenoids of each genus are included in our Supplementary File 1 Table S2. The sesquiterpenoids contained in the references you provided us are also included.
Uncovering hidden sesquiterpene biosynthetic pathway through expression boost area-mediated productivity enhancement in basidiomycete. Asai S, Tsunematsu Y, Masuya T, Otaka J, Osada H, Watanabe K. J Antibiot (Tokyo). 2020;73(10):721-728. doi: 10.1038/s41429-020-0355-9.
Isolation of a gene cluster from Armillaria gallica for the synthesis of armillyl orsellinate-type sesquiterpenoids. Engels B, Heinig U, McElroy C, Meusinger R, Grothe T, Stadler M, Jennewein S. Appl Microbiol Biotechnol. 2021;105(1):211-224. doi: 10.1007/s00253-020-11006-y.
Response: The sesquiterpene synthases Pro1 and Cop6 identified in the two papers you cited are included in our article. In addition, we have added to the existing ones the sesquiterpene synthases identified in the recent papers.
53 Wang, Q.; Liu, J.-K.; Zhao, Q.; He, Q.-L. Mechanistic Investigations of Hirsutene Biosynthesis Catalyzed by a Chimeric Sesquiterpene Synthase from Steccherinum ochraceum. Fungal Genetics and Biology 2022, 161, 103700, doi:10.1016/j.fgb.2022.103700.
The file “jof-1777288-supplementary” should be converted from an excel sheet (there are no calculations anyway!) into a word or pdf document instead of presenting 18 empty columns. Typically, sequence data are presented in lines, in the case of proteins with the amino end as the start on the left. If several proteins are presented, they appear one below the other to better indicate consensus motifs (if any). Gene bank numbers may appear at the very left margin of such a comparative picture, and references may be put into the figure legend. This is common standard:
|
# |
STC4 |
MVQFRIPDLLSCLPACIKATNADNDILQAGLVVIDQCHLTDHYKKD …. |
|
# |
STC9 |
MFRFDHPSSFILQNICDITGAVFELKENPLREQANTAVLKWFKGFN …. |
|
# |
STC 15 |
MSAATSQLLPSALATKIILPDLVAHCDFTLRYNRHRKQITRETKRWLF …. |
|
# |
Cop1 |
MVNLSWYWQGQGNISKSIGPPSQTYTKSVLREQSMTFRMLALQSGL …. etc |
Here, the lack of consensus motifs becomes quite obvious, a problem in primer design and heterologous production of STS.
Response: We have modified the format of our included amino acid sequences according to your suggestion, and sequences with JGI ID or Genbank ID are marked in the gene name, and we have modified all sequences to FASTA format and submitted them as PDF files (Supplementary File 2).
Reviewer 3 Report
Sesquiterpenes are important natural products with a variety of biological activities. In this review, Wu et al. summarized the sesquiterpenoids from Basidiomycetes and from heterologous expression of STS genes.
Major comments:
1, Why does Basidiomycota produce more sesquiterpenes? Please discuss it based on the living environment and the life cycle of Basidiomycota.
2, How many sesquiterpenoids have been identified in basidiomycetes? How many sesquiterpenoids have been identified through heterologous expression? The manuscript should be reorganized.
3, How many species have been sequenced in Basidiomycota? From the data base, how many STS genes have been predicted?
Minor comments:
1, Gene name should be italic.
2, In Abstract, “their biosynthesis is regulated by sesquiterpene synthase (STS)” should be “they are biosynthesized through the sesquiterpene synthase (STS)”.
Author Response
Sesquiterpenes are important natural products with a variety of biological activities. In this review, Wu et al. summarized the sesquiterpenoids from Basidiomycetes and from heterologous expression of STS genes.
Response: Thank you for your comments on this review, which lead us to improve our work. We have responded to your comments point by point, and we have also revised the main text of the manuscript.
Major comments:
- Why does Basidiomycota produce more sesquiterpenes? Please discuss it based on the living environment and the life cycle of Basidiomycota.
Response: As suggested, We added a new paragraph in the conclusion.
“Sesquiterpenes play a very important role in Basidiomycetes. Sesquiterpenes can attract insects for pollination [1], defend against other organisms or parasites [2], and play an important role in Basidiomycetes physiological effect. Also, when Basidiomycetes form a symbiotic relationship with plants. The sesquiterpenoids produced by Basidiomycetes can act as phytohormones [3,4]. Therefore, Basidiomycetes produce many different kinds of sesquiterpenes to help them better adapt to the living environment.
We mentioned the influence of the external environment on the production of sesquiterpenes by Basidiomycetes, including external physical factors (light, temperature, etc.) and chemical factors (exogenous chemical substances, etc.). The changes of sesquiterpenes during the development of basidiomycetes and their biological roles are poorly understood. At present, there are studies on the changes of sesquiterpenes during the development of the fruiting bodies of the Cyclocybe aegerita AAE-3 strain. The development of the fruiting body changes, and greater changes occurred during the sporulation process. In the early stage of sporulation, mainly alcohols and ketones, and in the later stage of sporulation, sesquiterpenes such as Δ6-protoilludene, α-cubebene and δ-cadinene appeared. substance. After sporulation, sesquiterpenoids decreased and other compounds appeared, mainly octan-3-one [5].”
- How many sesquiterpenoids have been identified in basidiomycetes? How many sesquiterpenoids have been identified through heterologous expression? The manuscript should be reorganized.
Response: According to our data, a total of 954 different sesquiterpenes from 116 genera have been identified in Streptomyces, with detailed information in Supplementary File 1 Table S2. After this revision, a total of 124 sesquiterpene synthases from Streptomyces are now included in our article, producing 49 different sesquiterpenoids.
- How many species have been sequenced in Basidiomycota? From the data base, how many STS genes have been predicted?
Response: A total of 629 genomes of Basidiomycete are included in the JGI database(Figure 1.), and a total of 792 genomes of Basidiomycete are included in the NCBI(Figure 2.). Because we can't download the full genome sequence, the data is not shown in the manuscript. Review paper by Schmidt-Dannert, C., the average number of sesquiterpene genes per strain in basidiomycetes is more than 12 (Reference 25).
- Schmidt-Dannert, C. Biosynthesis of Terpenoid Natural Products in Fungi. In Biotechnology of Isoprenoids; Schrader, J., Bohlmann, J., Eds.; Advances in Biochemical Engineering/Biotechnology; Springer International Publishing: Cham, 2014; Vol. 148, pp. 19–61 ISBN 978-3-319-20106-1.
Figure 1. Number of genomes of Streptomyces included in JGI database
Figure 2. Number of genomes of Streptomyces included in NCBI database.
Minor comments:
- Gene name should be italic.
Response: We have made changes in the article as you suggested.
- In Abstract, “their biosynthesis is regulated by sesquiterpene synthase (STS)” should be “they are biosynthesized through the sesquiterpene synthase (STS)”.
Response: We have made changes to the article as you suggested.
[1] de Bruyne, M.; Baker, T.C. Odor Detection in Insects: Volatile Codes. J Chem Ecol 2008, 34, 882-897, doi:10.1007/s10886-008-9485-4.
[2] Kramer, R.; Abraham, W.-R. Volatile Sesquiterpenes from Fungi: What Are They Good For? Phytochem Rev 2012, 11, 15-37, doi:10.1007/ s11101-011-9216-2.
[3] Plett, J.M.; Martin, F. Blurred Boundaries: Lifestyle Lessons from Ectomycorrhizal Fungal Genomes. trends in genetics 2011, 27, 14- 22, doi:10.1016/j.tig.2010.10.005.
[4] Wu, J.; Kawagishi, H. Plant Growth Regulators from Mushrooms. j Antibiot 2020, 73, 657-665, doi:10.1038/s41429-020-0352-z.
[5] Orban, A.; Hennicke, F.; Rühl, M. Volatilomes of Cyclocybe Aegerita during Different Stages of Monokaryotic and Dikaryotic Fruiting. Biological Chemistry 2020, 401, 995–1004, doi:10.1515/hsz-2019-0392.
Reviewer 4 Report
The article describes the diversity of basidiomycete fungi with ability to product sesquiterpenes.
The article is well organized and well presented. But several sentences in the article looks incomplete (eg. lines 21 and 22).
Though the major part of information presented is talking about the genome studies with reference to the enzymes produced by several of the basidiomycete fungi for sesquiterpenes, some connecting link between the enzymes produced, their diversity in terms of structure and activity need to be included to make this review complete. Currently the article is little biased, talks much about the enzymes but based on genome sequence data.
There is list of organisms and enzymes produced by the different fungi for sesquiterpene production is listed with pathways, but comparison of these with highlights on similarities and differences will be required to be included. Additional details on enzymes involved and their characterization might add value.
Author Response
The article describes the diversity of basidiomycete fungi with ability to product sesquiterpenes.
Response: Thank you for your comments on our review. as these comments led us to an improvement of the work.
The article is well organized and well presented. But several sentences in the article looks incomplete (eg. lines 21 and 22).
Response: Thank you very much for your careful review, we have revised the article, made corrections to incomplete or illogical sentences, and reflected it in the manuscript in revision mode.
Though the major part of information presented is talking about the genome studies with reference to the enzymes produced by several of the basidiomycete fungi for sesquiterpenes, some connecting link between the enzymes produced, their diversity in terms of structure and activity need to be included to make this review complete. Currently the article is little biased, talks much about the enzymes but based on genome sequence data.
There is list of organisms and enzymes produced by the different fungi for sesquiterpene production is listed with pathways, but comparison of these with highlights on similarities and differences will be required to be included. Additional details on enzymes involved and their characterization might add value.
Response: We mentioned some common features of Basidiomycete sesquiterpene synthases in the original text. According to your suggestion, we have added the partial sequences of Cop3, 4, 6, STC4, 9, and 15 in the manuscript. In comparison, these enzymes have been tested for the effect of point mutation on enzyme activity, we summarize the functional effects of point mutations in these enzymes. And add this part in the discussion section.
Site-specific mutations are tools to study enzyme structure, function, catalytic mechanism. Which include single and combinatorial mutations [1]. In the study of sesquiterpene synthases of basidiomycetes, point mutations at residues near the conserved region are mostly used. The current point mutation experiments for sesquiterpene synthase of basidiomycetes are concentrated near the conserved regions of RY Pair and NSE Triad (Figure 5.).
Figure 5. (a) Comparison of the conserved regions of H-α1 loop and NSE/DTE motif. (b) Comparison of conserved regions of the RY Pair to the Thirteen Positions Upstream of the RY Pair. The MUSCLE algorithm was used to compare the following protein sequences: Cop3 (XP_001832925), Cop4 (XP_001836356), Cop6 (XP_001832549) in Coprinopsis cinerea; STC4 (KAH0582448), STC9 (KAH0583476), STC15 (KAG5341349) in Macrolepiota albuminosa.
Cop3, Cop4, and Cop6 were experimentally point mutated at the sites of their H-α1 loops, respectively. K233 in Cop4 and K251 in Cop3 did not play a major role in the side chain and ligand interaction network formed during active-site closure; mutations in Cop4 (K233, H235, T236, N238, and N239) showed that mutations in the H-α1 loop region site significantly altered the type of product, with the mutation of N239L had the greatest effect on the product; the mutation of Cop6 (C236, E237,and N240) showed that mutation of the H-α1 loop region site did not alter the product of Cop6. Structural modeling of Cop enzyme pointed to a potential interaction between the H-α1 loop and conserved residues of two metal-binding motifs (DDXXDD and NSE/DTE). Potential interactions between the conserved Asp/Glu and Arg, Asn and Lys sites in some sesquiterpene synthases in several fungal and plant may stabilize the closed enzyme conformation by closing the H-α1 loop [2]. STC4 was transformed into germacrene A (4) synthase after single site W335F mutation; various variants of the W314 point mutation in STC15 were unable to obtain expressed protein, and enzyme activity was reduced after mutation of the putative C311 active site in STC9 [3]. A triple mutant Cop2(17H2) was obtained by error-prone PCR, Cop2(17H2) contains both three mutations in L59H, T65A and S310Y, the three mutations make Cop2(17H2) products tend to be specific, and compared to the original Cop2 Cop2(17H2) is more inclined to produce Germacrene D-4-ol (2) [4].
- Yang, H.; Li, J.; Du, G.; Liu, L. Microbial Production and Molecular Engineering of Industrial Enzymes. in Biotechnology of Microbial Enzymes; Elsevier, 2017; pp. 151-165 ISBN 978-0-12-803725-6.
- López-Gallego, F.; Wawrzyn, GraysonT.; Schmidt-Dannert, C. Selectivity of Fungal Sesquiterpene Synthases: Role of the Active Site's H-1α Loop in Catalysis. Appl Environ Microbiol 2010, 76, 7723-7733, doi:10.1128/AEM.01811-10.
- Burkhardt, I.; Kreuzenbeck, N.B.; Beemelmanns, C.; Dickschat, J.S. Mechanistic Characterization of Three Sesquiterpene Synthases from the Termite-Associated Fungus Termitomyces. Org. Biomol. Chem. 2019, 17, 3348-3355, doi:10.1039/C8OB02744G.
4. Lauchli, R.; Pitzer, J.; Kitto, R.Z.; Kalbarczyk, K.Z.; Rabe, K.S. Improved Selectivity of an Engineered Multi-Product Terpene Synthase. org. Biomol. Chem. 2014, 12, 4013-4020, doi:10.1039/C4OB00479E.
Round 2
Reviewer 3 Report
The authors have been improved this manuscript as suggested, I have no comment.